# Predicting adherence to gamified cognitive training using early phase game performance data: Towards a just-in-time adherence promotion strategy

Yuanying Pang[1], Ankita Singh[2], Shayok Chakraborty[2], Neil Charness[3], Walter R. Boot[3], Zhe He[1] *

1 School of Information, Florida State University, Tallahassee, Florida, United States of America,
2 Department of Computer Science, Florida State University, Tallahassee, Florida, United States of America,
3 Department of Psychology, Florida State University, Tallahassee, Florida, United States of America

* zhe@fsu.edu

## Abstract

### Background and objectives

This study aims to develop a machine learning-based approach to predict adherence to gamified cognitive training using a variety of baseline measures (demographic, attitudinal, and cognitive abilities) as well as game performance data. We aimed to: (1) identify the cognitive games with the strongest adherence prediction and their key performance indicators; (2) compare baseline characteristics and game performance indicators for adherence prediction, and (3) test ensemble models that use baseline characteristics and game performance data to predict adherence over ten weeks.

### Research design and method

Using machine learning algorithms including logistic regression, ridge regression, support vector machines, classification trees, and random forests, we predicted adherence from weeks 3 to 12. Predictors included game performance metrics in the first two weeks and baseline measures. These models' robustness and generalizability were tested through five-fold cross-validation.

### Results

The findings indicated that game performance measures were superior to baseline characteristics in predicting adherence. Notably, the games "Supply Run," "Ante Up," and "Sentry Duty" emerged as significant adherence predictors. Key performance indicators included the highest level achieved, total game sessions played, and overall gameplay proportion. A notable finding was the negative correlation between initial high achievement levels and sustained adherence, suggesting that maintaining a balanced difficulty level is crucial for long-term engagement. Conversely, a positive correlation between the number of sessions played and adherence highlighted the importance of early active involvement.

**Data Availability Statement:** The source code and data used for data analysis and model implementation can be accessed at the following

GitHub repository: https://github.com/
YuanyingPang/APPT_Game_Performance_
Analysis.

**Funding:** This work was supported by the National
Institute on Aging grant R01AG064529. This study
was also partially supported by University of
Florida-Florida State University Clinical and
Translational Science Award funded by National
Center for Advancing Translational Sciences under
Award Number ULITR001427. The funders had no
role in study design, data collection and analysis,
decision to publish, or preparation of the
manuscript.

**Competing interests:** The authors have declared
that no competing interests exist.

## Discussion and implications

The insights from this research inform just-in-time strategies to promote adherence to cognitive training programs, catering to the needs and abilities of the aging population. It also underscores the potential of tailored, gamified interventions to foster long-term adherence to cognitive training.

## Introduction

Worldwide, there were 727 million people 65 years of age or older in 2020 [1]. By 2050, this figure will be more than doubled to over 1.5 billion people, indicating that the world is going through a significant demographic change [1]. As people age, they are more likely to experience chronic diseases, which can have significant impacts on their quality of life and their ability to live independently. Mild cognitive impairment (MCI), Alzheimer's disease (AD), and related dementias are three of the most common chronic illnesses that lead to memory loss and deterioration of cognitive skills [2]. Even for older adults without MCI, AD, or dementia, their cognitive abilities may decline as they age [3]. Thus, there is a growing need for care services to support older adults, including medical care, long-term care, and social services. At present, the treatment of memory loss is mainly divided into pharmacological therapy and non-pharmacological therapy. While medications like cholinesterase inhibitors and memantine may help delay cognitive decline in some individuals with Alzheimer's disease, they are not effective for everyone [4, 5]. Additionally, even for those who do experience benefits, the effects may be modest and short-lived [5]. Expectedly, these modification treatments have been associated with side effects, including gastrointestinal symptoms and dizziness, which have often led to treatment discontinuation [4, 5]. Recently approved drugs for early-stage dementia such as donanemab-azt and lecanemab-irmb target the amyloid plaques that are considered a biomarker of MCI and Alzheimer's disease [6, 7]. Although successful at clearing amyloid, the drugs have limited success in slowing cognitive decline and are associated with severe side effects such as brain swelling [6, 7].

Non-pharmacological interventions represent another approach to delay memory loss in older adults [8]. Gamified cognitive training, which is a form of non-pharmacological intervention, has gained popularity over the last few year [9].For older people, gamified cognitive training has several potential benefits: offering a more engaging and motivating setting, increasing adherence and motivation, increasing effectiveness [10, 11]. Gamified cognitive training can also provide a more customized and enjoyable experience, which could increase its overall effect [11]. Nevertheless, some researchers found that gamified cognitive training yielded disparate outcomes with respect to enhancing memory, and there needs more studies to explore the different effects of different games [12].

We found many articles indicated that gamified environment can help older adults engage with long-term online intervention [13, 14]. But researchers also suggested that the training should be personalized to maximize adherence so that their effect can be shown [13]. The objective of the "Adherence Promotion with Person-Centered Technology" (APPT) project is to develop an adaptive reminder system to promote older adults' adherence to mobile-based gamified cognitive training, with the ultimate goal of promoting early detection and prevention of age-related cognitive decline [15]. In previous studies, we developed machine learning models to predict participants' adherence at different levels (overall and weekly) using data collected from a previous cognitive training intervention, using a variety of baseline measures

(demographic, attitudinal, and cognitive ability variables), as well as deep learning models to predict the following week's adherence using variables derived from the previous week's training data [16]. This study revealed that both individual differences and previous intervention interactions provide valuable information for predicting adherence, and these insights can provide initial hints regarding who to target with adherence support strategies and when to provide support [16]. However, the overall adherence throughout the training phase was only predicted using baseline measures with a moderate prediction accuracy (AUROC = 0.71) with the best performing machine learning model). Another study by the APPT team, used multivariate time series data from the game training data consisting of time-dependent variables (i.e., duration for which a participant played, number of sessions, max level reached, and the number of games performed) to assess whether participants would meet the minimum adherence criteria on day N+1 given their previous N-day continuous play pattern. These two studies present the first steps toward the development of an innovative AI-based reminder system to promote adherence to mobile-based cognitive training program [17].

In this paper, we investigated the predictive power of participants' performance on different cognitive games during the initial two weeks of training and their subsequent 10-week adherence to the computerized cognitive training program. This approach helps in identifying consistent adherence patterns, which is crucial for developing robust, just-in-time intervention strategies. While daily predictions might offer more granular insights, the 10-week average offers a broader perspective that informs sustained engagement and practical intervention planning. Further, the development of a just-in-time adherence support system that capitalizes on both short-term (day-to-day) and long-term (overall adherence over the remainder of the intervention) predictions would be more effective than a system that relies solely on short-term predictions. Specifically, the reminder system can be fine-tuned based on the combined forecasts. For example, if the long-term model predicts a high risk of dropout and the short-term model indicates a high probability of non-engagement over the next few days, this might present a more serious concern compared to scenarios where the short-term model predicts an adherence lapse but the long-term model predicts overall high adherence for the remainder of the trial. In such cases, the system might deploy more engaging, context-aware reminders that address specific barriers to long-term adherence to reduce the risk of attrition while also supporting short-term reengagement. We hypothesize that performance on different cognitive games during the initial two weeks of cognitive training is predictive of the subsequent 10-week adherence. Additionally, we conduct a comparative analysis between the predictive capabilities of baseline factors and game performance metrics. We also examine the performance of ensemble models that incorporate both baseline factors and game performance data in predicting adherence. Furthermore, we investigate which specific games possess stronger predictive power regarding adherence and identify key performance indicators within these games that drive this predictive ability. A study by Turunen et al. used various cognitive, demographic, lifestyle, and health-related variables to predict older adults' adherence to computer-based cognitive training [18]. Their analysis included both bivariate and multivariate techniques, utilizing a Zero-Inflated Negative Binomial (ZINB) model to predict adherence. In contrast, our study incorporates not only these baseline predictors but also game performance data. This comprehensive analysis provides more robust and reasonable results, which can be used to personalize cognitive training programs and improve adherence among older adults. The insights obtained from this research could aid in tailoring adherence promotion strategies to support individuals who may need extra support to maintain adherence to cognitive training. Moreover, the identification of the most influential factors associated with adherence can facilitate the design of personalized and effective cognitive training programs that cater to the unique needs and abilities of the target population.

**Table 1. Descriptive statistics of the clinical trial data used in this study.**

| | Age | Gender | | | Text Message Reminder Type | | |
|---|---|---|---|---|---|---|---|
| | Mean (SD) | Count (Percent%) | | | Count (Percent%) | | |
| | | Female | Male | Unknown | Negative-Framed | No Message | Positive-Framed |
| Minimal Adherence | | | | | | | |
| 0 | 72.9(5.7) | 38(48.7) | 21(55.3) | 1(50) | 24 | 18 | 18 |
| 1 | 72.4(5.4) | 40(51.3) | 17(44.7) | 1(50) | 14 | 20 | 24 |
| Full Adherence | | | | | | | |
| 0 | 72.9(5.7) | 39(50) | 23(60.5) | 1(50) | 26 | 19 | 18 |
| 1 | 72.3(5.4) | 39(50) | 15(39.5) | 1(50) | 12 | 19 | 24 |

## Materials and methods

### The cognitive training trial

The dataset was obtained from a prior clinical trial conducted on the Mind Frontiers mobile-based cognitive training game suite, designed explore how different message framings (positive-framed and negative-framed) about brain health influence adherence to a technology-based cognitive intervention and to identify individual differences that predict adherence [19]. Florida State University Institutional Review Board approved the study protocol (IRB #: 2017.20622) and informed consent form. The recruitment period of this study was between July 1, 2017, and March 1, 2018. All the recruited participants provided written consent with their signatures. No minors were included in the study. The trial recruited 118 older adults living in the community, with a mean age of 72.6 years and a standard deviation of 5.54. Among these older adults, 78 are female (66%), 38 are male (32%), and 2 are unknown (2%) (see Table 1). Participants were randomly assigned to one of three groups: no message, positive-framed messages, or negative-framed messages (see Table 1). Those in the positive-framing group received messages emphasizing the benefits of engaging with cognitive training (e.g., "Regular metal challenge can have a positive impact on the brain"), while those in the negative-framing group received messages highlighting the risks of not engaging in such training (e.g., "Infrequent mental challenge can have a negative impact on brain") [19]. Participants were instructed to engage in the cognitive training program comprising seven separate games (see Table 2) for five days a week for 45 minutes each session. The collected data includes the levels achieved in the games, ranging from 1 to 58, as well as five possible outcomes: Defeat, Stalemate, Victory, Abort, and Not Yet Finished. Two phases make up the dataset used in this research [19]. Participants in Phase 1 were required to adhere to a strict timetable for 12 weeks, playing for 45 minutes a day, 5 days in the week. The same participants in Phase 2 were asked to play as frequently as they desired during this unstructured, 6-week period.

### Defining adherence and exploring predictive factors

As adherence cannot be determined in the absence of a prescribed schedule, we only examined the data gathered during the structured phase (Phase 1). Based on daily and weekly training interactions, quantitative and categorical weekly adherence at the minimum and full levels were established for the structured phase. Weekly adherence at a minimal level was defined as the number of days individuals met the minimum requirement in a day ($> = 10$ minutes) divided by five; weekly adherence at a full level ($> = 36$ minutes, representing 80 percent of 45 min) was defined as the number of days individuals met the full requirement in a week divided by five. The subsequent 10-week adherence at the minimum level and full level are the average weekly adherence for the entire 10 weeks, respectively. Following the definition of minimal

**Table 2. Games in mind frontier.**

| Game | Details | Training Objectives |
|---|---|---|
| Ante Up (AU) | Players are shown cards organized in a certain pattern and must replicate this pattern over the course of a specified number of moves with the cards they are provided. | Logic (planning) game |
| Irrigator (IR) | Players are tasked with building a water pipeline from a well to various targets before time runs out using provided pipe pieces that change with each turn. As players progress, various obstacles must be avoided to reach the target. | Spatial cognition game |
| Pen 'Em Up (PEU) | Players must sort objects dropped from a UFO into two pens by swiping either left or right based on specific criteria provided at the start. The sorting criteria varies based upon the objects' characteristics (e.g., trees, farm animals) or style (e.g., plain, striped). | Sorting and rule memory game |
| Riding Shotgun (RS) | Players are riding in a horse drawn wagon, and the scene in front of the wagon contains a grid of tiles that could light up one at a time. The player must remember the sequence in which tiles of the grid are illuminated. They must then replicate the pattern in the correct order. | Memory and recall game |
| Trader Jack's (TJ) | Players are tasked with choosing an item or set of items that would be equal in value to items on a scale needing to be balanced. | Logic (reasoning) game |
| Sentry Duty (SD) | Players are tasked with remembering the sequence in which sentries outside of a fort wall lift a lantern and say a word. They must decide whether the location and word of the current sentry matches that of the sentry N turns prior. | Memory and recall game |
| Supply Run (SR) | Players adopt the role of a merchant traveling through a town. Townspeople request items along the way. The player must remember the last item requested from each of the provided categories so they may be purchased at a town store at the end of the trip. | Memory updating game |

and full-level adherence, participants were divided into two groups using the median value of each.

We collected comprehensive game data from every participant during their adherence with these cognitive training games, encompassing the number of sessions engaged in, the outcome of each play, and the level achieved in each play. The data presented in Fig 1 offers a preliminary summary of the mean frequency at which individuals engaged in each of the seven games over the initial two-week and twelve-week periods. Additionally, it illustrates the proportion of various outcomes observed throughout the same time intervals. The difference in play count

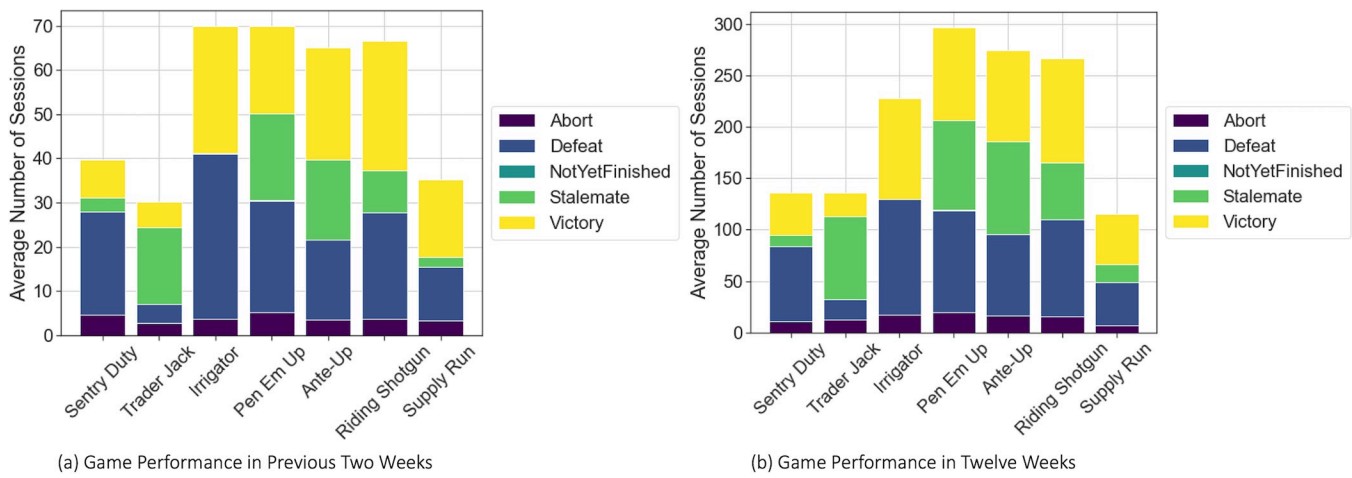

(a) Game Performance in Previous Two Weeks          (b) Game Performance in Twelve Weeks

**Fig 1.** Average sessions of different games in previous two weeks (a) and in whole trial (b).

may be due to differences in the duration of the game, which can affect the overall number of plays. Therefore, using play frequency as a metric for assessment may not be sufficient for a direct comparison. Analyzing the win-loss ratios provides valuable information on the performance patterns within each game. By comparing the win-loss percentages of each game during the early two weeks with the game performance across a whole twelve-week span, discernible patterns become evident. Significantly, the game "Trader Jack" demonstrates a greater percentage of stalemates compared to other games, a pattern that remains persistent during both the initial two weeks and the full twelve-week period. In contrast, the victory rates for games like "Riding Shotgun" and "Ante Up" are higher in the first two weeks compared to the twelve-week timeframe, suggesting a variation in performance with time. S2 Fig. displays an area chart illustrating the fluctuations in the number of results per game per week over a span of 12 weeks. The x-axis shows a time frame of one to twelve weeks, while the y-axis shows the total number of game outcomes every week. Each hue symbolizes a specific outcome in the game. The results consistently align with the overall win-loss ratio.

We developed machine learning models for each game, and based on these game performance data we created some predictors including the number of sessions they played for each of the 7 games in the first two weeks, the proportion of each of the 7 games played in terms of the number of sessions in the first two weeks, the proportion of each outcome for each game over all the sessions on the same game the first two weeks, the highest level they reached in initial two weeks, and how many days a person spent on reaching the median level of each game in the first two weeks. The original data variable information and the calculation methods of these three categories of variables are shown in Table 3 below. In defining the variable

**Table 3. Data obtained through gamified cognitive training interactions.**

| Category | Details |
|---|---|
| Text message reminder type | Experimental condition assigned to participants: those receiving positive-frame messages were coded as 1, those receiving negative-frame messages were coded as -1, and those receiving no message were coded as 0. |
| Sessions | Sessions initiated by participants |
| Game levels | Levels for games with maximum levels ranging from 16 to 58 for different games |
| Game outcomes | Outcome of each game. Total 5 possible outcomes (Defeat, Stalemate, Victory, Abort, Not Yet Finished) |
| Number of sessions | The total number of sessions of different games participants played in previous two weeks |
| Percentage of games | The number of sessions for each game divides by the total number of sessions they played in previous two weeks |
| Percentage of outcomes | The number of sessions resulting in each specific outcome divided by the aggregate number of sessions conducted for each game in previous two weeks. |
| Maximum level | The highest-level participants reached in previous two weeks |
| Number of days reaching middle level group | The number of days they spent on reaching median level for each game in previous two weeks |
| Minimal weekly adherence | The number of days they played at least 10 mins divided by 5 |
| Full weekly adherence | The number of days they played at least 36 mins divided by 5 |
| 10-week minimal adherence | Average of minimal weekly adherence from week 3 to week 12 |
| 10-week full adherence | Average of full weekly adherence from week 3 to week 12 |
| Minimal adherence level | If participants' 10-week minimal adherence value is greater than the median 10-week minimal adherence for all participants, it will be recorded as "1". Otherwise, it will be recorded as "0". |
| Full adherence level | If participants' 10-week full adherence value is greater than the median 10-week full adherence for all participants, it will be recorded as "1". Otherwise, it will be recorded as "0". |

"number of days reaching middle level group," we categorized participants' maximum levels from various games over the initial two weeks into three distinct level groups: low, middle, and high. These levels were determined using the 25th and 75th percentiles as thresholds. Specifically, levels falling within the 0-25th percentile were classified as "low level group", those within the 25th-75th percentile as "middle level group", and scores exceeding the 75th percentile as "high level group". "Number of days reaching middle level group" represents the number of days within a two-week span that a participant reaches this middle level group. If a participant does not achieve the middle level group within these two weeks, this variable is set to 15 days.

Prior to the beginning of this program, a survey was administered to gather participants' baseline information, encompassing data on demographics, attitudes, and cognitive abilities. For a comprehensive overview of attitude and cognitive scores for all participants, please refer to S1 Table in the Support Information. The measurements of attitudinal and cognitive assessments comprise composite scores that assess technical proficiency, self-efficacy, subjective cognition, perceived benefits, objective reasoning, objective processing speed, objective memory instant recall, and objective memory delayed recall. These composite scores are computed using numerous z-scores, as shown in Table 4.

To assess the relationship between both continuous and categorical variables with adherence outcomes, t-tests were conducted for continuous variables to determine if their mean values differed significantly between participants who met the adherence threshold and those who did not. Additionally, Chi-square tests were applied to categorical variables to evaluate whether significant associations existed between the categorical variables and adherence. These analyses helped identify key predictors associated with adherence based on both continuous and categorical predictors. We chose those measures that showed statistical significance (alpha = 0.1) in the prediction models, and we ran separate analysis on both minimal level adherence and full level adherence to choose different sets of predictors. To mitigate the potential reduction in validity due to the small sample size in the "Unknown" gender category, we consolidated the gender variable into two groups: "Female" and "Male and Unknown." This adjustment allowed us to apply the Chi-Square test. This choice is also consistent with our

**Table 4. Calculation of composite scores using z-scores.**

| Composite Score | Measurement | Mean (SD) |
| --- | --- | --- |
| Composite score for technology proficiency | Average z-scores for the Computer Proficiency Questionnaire [20], Mobile Device Proficiency Questionnaire (MDPQ) [21] | 2.619 (0.772) |
| Composite score for self-efficacy | Averaging the z-scores of the General Self-efficacy Questionnaire (GSE) [22] and the Technology Self-efficacy Questionnaire (TSE) [22] | 0.000 (0.800) |
| Composite score for subjective cognition | Average z-score for the Instrumental Activities of Daily Living survey (IADL) [23], Perceived Deficits Questionnaire (PDQ) [24], and Memory Self-Efficacy Questionnaire (MSEQ) [25] | -0.010 (0.644) |
| Composite score for perceived benefits | Average z-score for Brain Training (INDP) [26] and Independence Survey and Brain Training Belief Scale (NICT) [27] | 0.063 (0.710) |
| Composite score for objective reasoning | Average z-scores for Raven's Advanced Progressive Matrices [28, 29] and Letter Sets [30] | -0.0133 (0.863) |
| Composite score for objective processing speed | Average the z-scores for the Useful Field of View test (UFOV) [31] and Digit Symbol Substitution [32] | -0.010 (0.644) |
| Composite score for objective memory immediate recall | Average z-score for Rey's Auditory Verbal Learning Test (Rey's AVLT) [33] and Hopkins Verbal Learning Test (Hopkins memory test, immediate recall) [34] | 0.068 (0.816) |
| Composite score for objective memory delayed recall | Average z-score for Rey's AVLT delayed recall [33] and Hopkins Verbal Learning test (Hopkins memory test, delayed recall) [34] | 0.007 (0.831) |

**Table 5. P-values between the variables and 10-week adherence at minimal and full levels.**

| | p-value | |
|---|---|---|
| | **Minimal Level** | **Full Level** |
| Age | 0.635 | 0.564 |
| Gender | 0.652 | 0.403 |
| Test Message Reminder Type | 0.169 | **0.064** |
| Z-score for UFOV task | 0.575 | 0.529 |
| Z-score for digit symbol | 0.774 | 0.606 |
| Z-score for Ravens | 0.970 | 0.659 |
| Z-score for Letter sets | 0.936 | 0.594 |
| Z-score for Hopkins memory test, immediate recall | 0.348 | 0.421 |
| Z-score for Hopkins memory test, delayed recall | **0.037** | **0.039** |
| Z-score for Rey's AVLT, immediate recall | 0.100 | **0.060** |
| Z-score for Rey's AVLT, delayed recall | **0.030** | **0.014** |
| Z-score for IADL survey | 0.975 | 0.936 |
| Z-score for Brain Training Independence measure | 0.804 | 0.764 |
| Z-score for NICT | 0.173 | 0.278 |
| Z-score for MSEQ | 0.587 | 0.699 |
| Z-score for Perceived Deficits questionnaire (PDQ) | 0.676 | 0.947 |
| Z-score for General Self-efficacy (GSE) | **0.005** | **0.010** |
| Z-score for Technology Self-efficacy (TSE) | **0.054** | **0.068** |
| Z-score for Computer Proficiency Questionnaire (CPQ) | 0.786 | 0.978 |
| Z-score for Mobile Device Proficiency Questionnaire (MDPQ) | 0.611 | 0.445 |
| Z-score for tech readiness questionnaire | **0.055** | **0.084** |
| Composite score for technology proficiency | 0.354 | 0.277 |
| Composite score for self-efficacy | **0.003** | **0.006** |
| Composite score for IADL | 0.547 | 0.406 |
| Composite score for reasoning | 0.703 | 0.339 |
| Composite score for subjective cognition ability | 0.643 | 0.842 |
| Composite score for objective cognition ability, processing speed | 0.605 | 0.485 |
| Composite score for objective cognition ability, immediate recall | 0.152 | 0.132 |
| Composite score for objective cognition ability, delayed recall | **0.020** | **0.013** |
| Composite objective cognition ability, immediate and delayed recall | **0.047** | **0.036** |

previous research [16]. The p-values for each profile factor are displayed in Table 5. For the prediction of 10-week adherence at a minimal level, a total of five z-score variables and three composite score variables were selected as predictors. The predictors of 10-week adherence at the full level were chosen to be the participants' test message reminder type, six z-score variables, and three composite score variables.

## Predicting adherence

This study is structured around three primary objectives: 1) To determine the cognitive games that exhibit the most robust predictive strength for adherence and to pinpoint the key performance indicators within these games that contribute to this predictive capacity. 2) To perform a comparative analysis to assess the relative predictive abilities of various profile factors versus game performance metrics in forecasting adherence. 3) To evaluate the effectiveness of ensemble models that integrate both profile factors and game performance data in predicting adherence over a subsequent ten-week period.

We employed diverse classification methods, including logistic regression, ridge regression, support vector machines, classification trees, and random forests, to predict the adherence from Week 3 to 12. The models were developed using two different sets of predictors. The first set included performance metrics from participants in seven cognitive games, with separate machine learning models for each game's specific performance features. The second set of variables included baseline features, where z-scores and composite scores were used separately to create distinct models. This allowed for a comprehensive evaluation of the predictive power of the profile factors. Data normalization procedures were implemented to standardize the game performance features, ensuring comparability across various metrics. Additionally, a five-fold cross-validation approach was employed to assess the classification outcomes, enhancing the validity and reliability of the predictive models.

The ensemble modeling is a machine learning technique that combines many models, known as estimators, to create predictions by allowing the models to vote for the predicted class. There are two categories of voting classifiers based on the method of voting: soft and hard voting [35], which are also called unanimous and majority voting [36]. In hard voting, the final class prediction is decided through a majority vote from many estimator models. Soft voting involves the calculation of the probability for each class by each estimator in order to make the final class prediction [35]. Previous research has demonstrated the benefits of hybrid ensemble classifiers and voting-based ensembles in improving prediction accuracy and model robustness [36, 37]. By employing a soft voting ensemble modeling technique, the game performance models ("Supply Run", "Ante-Up", and "Sentry Duty") that exhibit higher predictability of adherence are combined with the baseline model (composite score model) that demonstrates higher predictability of adherence to forecast minimal adherence and full adherence. The homogeneous ensemble technique employs logistic regression as the base learner and aggregates predictions from individual models through a soft voting mechanism to enhance prediction accuracy and reduce variances and biases seen in individual models.

We utilized accuracy and Area Under the Receiver Operating Characteristic (AUROC) to evaluate several classification models and visualized them by line and radar charts. The AUROC is a widely used metric for evaluating the performance of classification models, particularly in binary classification tasks [38]. To elucidate which independent variables derived from game performance most effectively predict adherence, we employed SHAP (SHapley Additive exPlanations) values within the logistic regression framework for each game. SHAP values, based on game theory's Shapley values, offering a rigorous and consistent approach to measure the contribution of each feature to the predictive model [39]. The source code and data used for data analysis and model implementation can be accessed at the following GitHub repository: https://github.com/YuanyingPang/APPT_Game_Performance_Analysis.

## Results

Fig 2 illustrates a line chart where the x-axis represents various machine learning models. Each data point on the chart corresponds to the combined average accuracy or AUROC score. The blue line represents the mean accuracy/AUROC score of various machine learning models applied to seven different cognitive games with the purpose of predicting adherence. For example, the starting point on this blue line of the right panel of Fig 2(A) is the average accuracy achieved by applying seven logistic regression models to their corresponding games. The orange line on the chart represents the accuracy/AUROC values obtained from the models with two different types of baseline predictors: composite score and z-scores. Based on Fig 2, we found that, on average, the features obtained from game performance are more reliable in predicting adherence compared to baseline features. Moreover, average accuracy and AUROC

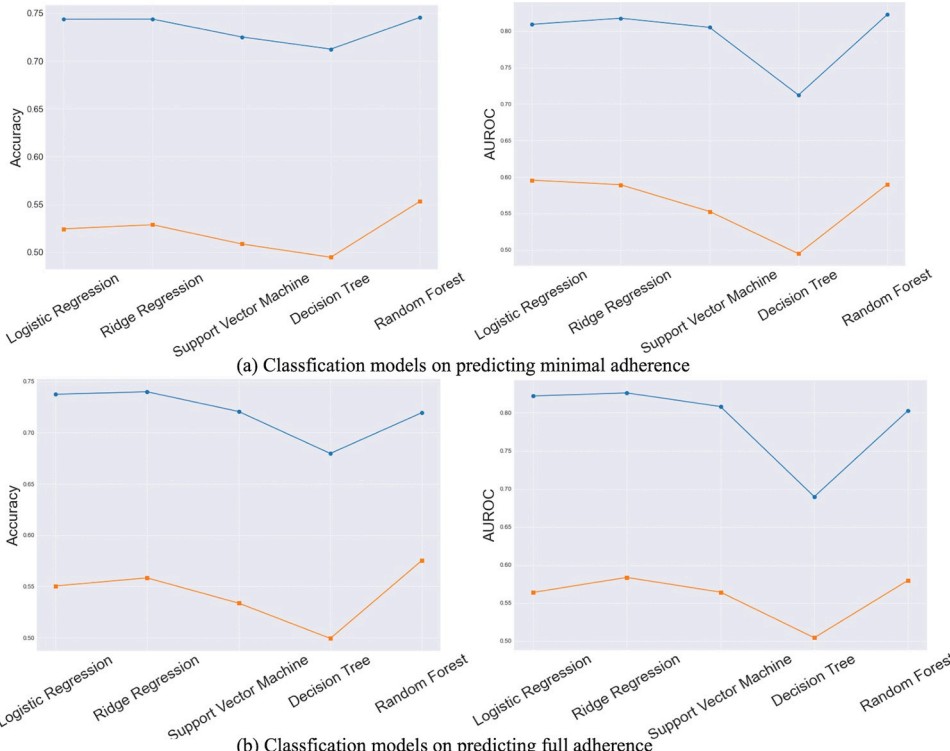

(a) Classfication models on predicting minimal adherence

(b) Classfication models on predicting full adherence

**Fig 2.** Line Graph Illustrating the Average Predictive Performance of Different Models for Minimal (a) and Full Adherence (b); Note: The blue line represents average model accuracy using game performance predictors. The orange line indicates accuracy/AUROC from models with baseline predictors: composite score and z-scores.

values of all machine learning models that used game performance features were higher than 0.7 when predicting minimal adherence. All models, except for the Decision Tree model, achieved average accuracy and AUROC scores above 0.7 when using game performance measures to predict full adherence. These metrics indicated a moderately strong ability of the models to classify participants based on their game performance in previous two weeks. Accuracy, representing the proportion of correctly predicted instances, suggests that our models can reliably identify adherence behaviors. The AUROC value, which measures the model's ability to distinguish between adherent and non-adherent participants, further supports the model's discriminative power. We also noted that logistic regression and ridge regression models regularly achieved better performance compared to other models, which supports our choice of logistic regression as the baseline model for further ensemble modeling.

To identify the cognitive game with the most robust predictive power for adherence, we visualized the performance metrics, as illustrated in Fig 3. In these radar charts, each axis represents a different machine learning model, and the distinct colors correspond to different games. The radar charts elucidate the differential predictive capabilities of each game across various models, highlighting their unique strengths in adherence forecasting. When synthesizing the predictive outcomes for both minimal and full adherence for each game, the aggregated accuracy rate consistently exceeds the 70% threshold. Notably, "Supply Run", "Ante Up", and "Sentry Duty" excelled in terms of predictive validity related to adherence levels. When using the performance of these three games in the first two weeks to predict the following 10-week adherence, both AUROC and AUPRC values were above 0.8, indicating that they were highly effective in identifying adherents or non-adherents. These games were thus strategically

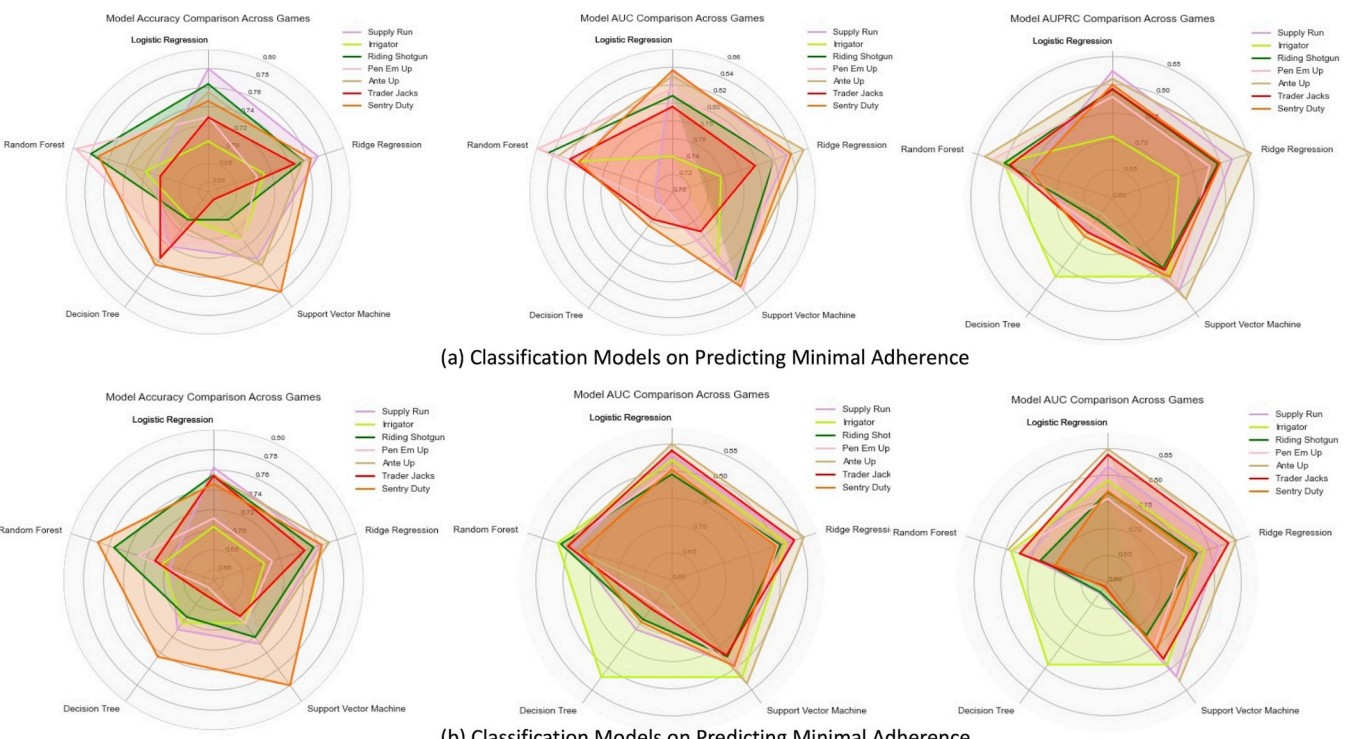

**Fig 3.** Comparative Analysis of Classification Models Using Game Performance Metrics to Predict Levels of Adherence: (a) Minimal Adherence, and (b) Full Adherence.

chosen to underpin the ensemble modeling process, aiming to enhance the accuracy of adherence predictions by integrating them with baseline feature data.

Table 6 presents the performance metrics of the ensemble model in comparison to the four logistic regression base models. The ensemble model, which synthesizes the predictive outputs from these base models, attained an AUROC of 0.86 and an AUPRC of 0.85 for predicting minimal adherence. Furthermore, for predicting full adherence, the ensemble model achieved an AUROC of 0.86 and an AUPRC of 0.86. These results signify an enhancement of over 0.03 for both metrics when contrasted with the most proficient individual base model. The table

**Table 6. Comparative performance metrics of base models and ensemble model.**

| Y = min adherence | | Accuracy | Precision | Recall | F1 score | AUROC | AUPRC |
|---|---|---|---|---|---|---|---|
| | X = Supply Run | 0.7801 | 0.7843 | 0.7801 | 0.7790 | 0.8378 | 0.8246 |
| | X = Ante Up | 0.7554 | 0.7577 | 0.7554 | 0.7552 | 0.8302 | 0.8108 |
| | X = Sentry Duty | 0.7460 | 0.7523 | 0.7460 | 0.7443 | 0.8365 | 0.8004 |
| | X = Composite score | 0.5417 | 0.5479 | 0.5417 | 0.5303 | 0.5995 | 0.6168 |
| | Ensemble Model | 0.7783 | 0.7868 | 0.7364 | 0.7540 | 0.8631 | 0.8506 |
| Y = full adherence | | Accuracy | Precision | Recall | F1 score | AUROC | AUPRC |
| | X = Supply Run | 0.7623 | 0.7712 | 0.7623 | 0.7601 | 0.8301 | 0.8168 |
| | X = Ante Up | 0.7547 | 0.7632 | 0.7547 | 0.7532 | 0.8502 | 0.8495 |
| | X = Sentry Duty | 0.7214 | 0.7231 | 0.7214 | 0.7205 | 0.8035 | 0.7703 |
| | X = Composite score | 0.5504 | 0.5532 | 0.5504 | 0.5429 | 0.5638 | 0.5980 |
| | Ensemble Model | 0.7457 | 0.7350 | 0.7272 | 0.7273 | 0.8642 | 0.8570 |

**Table 7. SHAP value analysis of game performance predictors across seven games.**

| Y = minimal adherence | Game | | | | | | | Avg Rank |
|---|---|---|---|---|---|---|---|---|
| Variables | Supply Run | Irrigator | Riding Shotgun | Pen 'Em Up | Ante-Up | Trader Jack's | Sentry Duty | |
| Number of sessions | 2 | 4 | 2 | 2 | 7 | 4 | 2 | 3.29 |
| Maximum level | 1 | 1 | 1 | 1 | 1 | 1 | 1 | 1.00 |
| Percentage of 'game' | 6 | 2 | 6 | 3 | 3 | 2 | 3 | 3.57 |
| Percentage of defeat | 5 | 7 | 4 | 8 | 6 | 8 | 7 | 6.43 |
| Percentage of abort | 9 | 3 | 7 | 5 | 9 | 7 | 5 | 6.43 |
| Percentage of victory | 7 | 6 | 5 | 7 | 5 | 6 | 9 | 6.43 |
| Percentage of stalemate | 4 | 9 | 9 | 4 | 4 | 5 | 4 | 5.57 |
| Percentage of NotYetFinished | 8 | 8 | 8 | 6 | 8 | 9 | 8 | 7.86 |
| Number of days reach median | 3 | 5 | 3 | 9 | 2 | 3 | 6 | 4.43 |
| Y = full adherence | Game | | | | | | | Avg Rank |
| Variables | Supply Run | Irrigator | Riding Shotgun | Pen 'Em Up | Ante-Up | Trader Jack's | Sentry Duty | |
| Number of sessions | 2 | 3 | 2 | 2 | 5 | 4 | 2 | 2.86 |
| Maximum level | 1 | 1 | 1 | 1 | 1 | 1 | 1 | 1.00 |
| Percentage of 'game' | 5 | 2 | 5 | 3 | 3 | 2 | 3 | 3.29 |
| Percentage of defeat | 8 | 5 | 6 | 8 | 6 | 7 | 7 | 6.71 |
| Percentage of abort | 6 | 4 | 8 | 7 | 9 | 9 | 4 | 6.71 |
| Percentage of victory | 7 | 8 | 3 | 5 | 7 | 6 | 9 | 6.43 |
| Percentage of stalemate | 4 | 9 | 4 | 4 | 4 | 8 | 6 | 5.57 |
| Percentage of not yet finished | 9 | 7 | 9 | 6 | 8 | 5 | 8 | 7.43 |
| Number of days reach median | 3 | 6 | 7 | 9 | 2 | 3 | 5 | 5.00 |

delineates the detailed performance metrics for each of the base models as well as for the ensemble model, are indicated in Table 6.

By calculating the SHAP value for each independent variable, we were able to ascertain its relative importance within the model. These values were then ranked for each variable across the logistic regression models corresponding to the seven games. Subsequently, we computed the average ranking of importance for each variable to derive a final hierarchy of variable importance, which is delineated in Table 7.

The aggregated SHAP value rankings reveal a consistent pattern across the models, indicating that the most salient predictors for predicting both minimal and full adherence are the same. Specifically, the variables that consistently emerged as the most influential are the highest level attained by the participants within the initial two weeks, the number of game sessions they played and the percentage of overall gameplay instances. These findings underscore the significance of early engagement and sustained interaction as key determinants of adherence.

To further investigate the relationship between individual game performance and subsequent 10-week adherence, we utilized the summary tables in S1 Fig. to demonstrate the effects of different game performances. The utilization of the SHAP values summary plot is a highly efficient method for evaluating the significance and effects of features within the framework of our predictive model. Each data point on the summary picture represents the SHAP value assigned to a particular feature or instance. For example, a feature characterized by mainly blue dots on the right side of the graph suggests that lower values of this variable enhance the probability of a positive outcome. We provide concise visual representations of the three most crucial game performance metrics, which are determined to be the top three based on the average SHAP values. We observed a negative correlation between the participants' maximum

level over the first two weeks and their likelihood of remaining engaged with the game. There was a positive correlation between the number of games played during the initial two weeks and the likelihood of continued adherence in cognitive training. Moreover, the percentage of different games played during the initial two weeks had a different influence on adherence. In the context of the game "Supply Run," there is a positive correlation between the percentage of games played and the likelihood of participants meeting both the minimal and full adherence standards. In the game "Riding Shotgun", a higher percentage correlated with a greater likelihood of achieving full adherence level. Other than the two previously mentioned, we observed a negative correlation between the percentage of the game played in the previous two weeks and the probability of achieving either minimal or full adherence.

## Discussion

Our research investigates the factors influencing overall adherence to cognitive training, incorporating baseline and game performance predictors. Our findings reveal that age and gender do not significantly affect adherence, which is consistent with prior research [18, 40]. Additionally, some researchers have suggested that memory function may impact adherence [18], which is aligned with our results. We found that delayed and immediate recall significantly affect overall adherence. However, some scholars argue that cognitive capability does not reliably predict commitment, persistence, or compliance with cognitive training [40]. Their study, however, only measured working memory and fluid reasoning as cognitive capabilities. Our study found that the composite score for objective reasoning also did not significantly affect adherence. Compared to the previously published study [16] using the same dataset, we found that by adding the game play data in the first two weeks, we can significantly improve the prediction accuracy of overall adherence even though the overall adherence is for 10 weeks, as opposed to 12 weeks used in the previous study [16].

Although personality predictors have some predictive ability regarding adherence, our primary finding is that game performance effectively predicts adherence to gamified cognitive training programs. The findings provide strong evidence that the game performance of specific cognitive games, namely "Supply Run," "Ante Up," and "Sentry Duty," have significant predictive ability for adherence levels. The performance measures, such as the maximum level achieved, the total number of game sessions played, and the proportion of overall gameplay instances, have been identified as important indicators of adherence. The mean accuracy and AUROC values consistently exceeding the 0.7 threshold across all machine learning models highlight the strength and reliability of game performance variables in predicting both minimal and full adherence. The radar maps clearly displayed the predictive strengths of each game, emphasizing their distinct contributions to adherence predictions. These strengths may stem from differences in game design. Previous research by Boot et al. found that game enjoyment and perceived challenge were related to motivation [41] and adherence, while Hu et al. highlighted the importance of enjoyment in encouraging sustained play in cognitive assessment games [42].

Although we did not measure participants' feelings about these games, we inferred some information from their early-stage game performance. The relationships between game performance and adherence rates are particularly noteworthy. The strong correlation between the number of sessions played and adherence suggests that early involvement is indicative of long-term adherence. The games in which participants played more sessions may offer greater enjoyment, leading to higher adherence. Conversely, the negative association between the highest level attained at the beginning of the program and continued adherence implies that maintaining an appropriate level of difficulty is crucial for long-term adherence. Some studies

have used Dynamic Difficulty Adjustment techniques to maintain difficulty levels and improve player engagement [43]. Future research could explore the use of such techniques to enhance adherence to cognitive training among older adults.

Furthermore, the combined ensemble model that incorporates both baseline characteristics and first two weeks' game performance data demonstrated superior performance compared to the individual logistic regression base models. This suggests that employing a comprehensive strategy that considers both individual characteristics and game-related data results in a more precise forecast of adherence. The findings have major implications for the development of tailored cognitive training programs. Through the identification of early indicators of potential lapses, interventions can be adjusted to boost user involvement and enhance adherence rates.

A few limitations should be noted to interpret the study more accurately. First, the sample size is relatively small and there were significantly more females than males in the study. This may impact the generalizability of the results. Nonetheless, we believe that the framework of using first two-week play data and baseline measures to predict adherence in the remaining weeks can be generalizable to other domains where a gamified intervention is used for treatment or behavioral change. Second, the game play data such as play time may not be accurate as participants may have put the tablet aside when the game was still running. Our recommendation for future studies is to use more advanced tracking technologies and improve user interfaces that can reduce errors in data capture.

## Conclusions

The findings of this study provide a significant contribution to the field of cognitive training by showing that game performance data can accurately predict user adherence. This predictive capability allows for the identification of individuals who are at risk and may need extra support to stay engaged in cognitive training programs. Although the study has uncovered encouraging results, it admits certain limitations. The research has not investigated the impact of various game settings on adherence to certain games, which is a potential field for future research. Furthermore, the study's emphasis on older adults may restrict the applicability of the findings to other demographic groups. Ultimately, this research establishes a basis for the creation of adaptive and individualized cognitive training programs. By utilizing the predictive capabilities of early game performance, we can potentially optimize the effectiveness of these programs, thereby promoting cognitive well-being in older adults. Future research should focus on improving the accuracy of prediction models and increasing their range of applications, to ensure that cognitive training continues to be a valuable and advantageous tool for a broader population.

## Supporting information

**S1 Fig.** Evaluation of Game Performance Effects Using SHAP Values Summary Plot (a) Minimal Adherence (b) Full Adherence.
(PDF)

**S2 Fig. Weekly fluctuations in game outcomes over 12 weeks.**
(PDF)

**S1 Table. Descriptive statistics of attitude and cognitive scores for all participants.**
(PDF)

## Author Contributions

**Conceptualization:** Neil Charness, Walter R. Boot, Zhe He.

**Formal analysis:** Yuanying Pang, Zhe He.

**Funding acquisition:** Walter R. Boot, Zhe He.

**Investigation:** Yuanying Pang, Zhe He.

**Methodology:** Yuanying Pang, Shayok Chakraborty, Neil Charness, Walter R. Boot, Zhe He.

**Project administration:** Zhe He.

**Validation:** Yuanying Pang, Zhe He.

**Writing – original draft:** Yuanying Pang, Zhe He.

**Writing – review & editing:** Yuanying Pang, Ankita Singh, Shayok Chakraborty, Neil Charness, Walter R. Boot, Zhe He.

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
