## [Decision Letter · Decision Letter 0]

27 May 2024

PONE-D-24-08688Predicting Adherence to Gamified Cognitive Training Using Early Phase Game Performance Data: Towards a Just-In-Time Adherence Promotion StrategyPLOS ONE

Dear Dr. He,

Thank you for submitting your manuscript to PLOS ONE. After careful consideration, we feel that it has merit but does not fully meet PLOS ONE’s publication criteria as it currently stands. Therefore, we invite you to submit a revised version of the manuscript that addresses the points raised during the review process.

The manuscript received mixed revision. I agree that more details and corrections are needed especially for what regards references, methodology and data analysis. I also agree that the main construct of adherence is not well defined and explained in the background and discussion. While it is important to predict adherence based on performance, one should not neglect important psychological predictors such as self-efficacy and (type of) motivation. Authors should expand on these aspects in background and discussion/limitations/future research directions. 

We look forward to receiving your revised manuscript.

Kind regards,

Stefano Triberti, Ph.D.

Academic Editor

PLOS ONE

Journal Requirements:

3. Thank you for stating the following financial disclosure: "This work was supported by the National Institute on Aging grant R01AG064529. This study was also partially supported by University of Florida-Florida State University Clinical and Translational Science Award funded by National Center for Advancing Translational Sciences under Award Number ULITR001427."  

4. Thank you for stating the following in the Acknowledgments Section of your manuscript: "This work was supported by the National Institute on Aging grant R01AG064529. This study was also partially supported by University of Florida-Florida State University Clinical and Translational Science Award funded by National Center for Advancing Translational Sciences under Award Number ULITR001427."

Please remove any funding-related text from the manuscript and let us know how you would like to update your Funding Statement. Currently, your Funding Statement reads as follows: "This work was supported by the National Institute on Aging grant R01AG064529. This study was also partially supported by University of Florida-Florida State University Clinical and Translational Science Award funded by National Center for Advancing Translational Sciences under Award Number ULITR001427."

5. We note that you have indicated that there are restrictions to data sharing for this study. For studies involving human research participant data or other sensitive data, we encourage authors to share de-identified or anonymized data. However, when data cannot be publicly shared for ethical reasons, we allow authors to make their data sets available upon request. For information on unacceptable data access restrictions, please see http://journals.plos.org/plosone/s/data-availability#loc-unacceptable-data-access-restrictions. 

Reviewers' comments:

Reviewer's Responses to Questions

**Comments to the Author**

1. Is the manuscript technically sound, and do the data support the conclusions?

Reviewer #1: Yes

Reviewer #2: No

2. Has the statistical analysis been performed appropriately and rigorously? 

Reviewer #1: Yes

Reviewer #2: I Don't Know

3. Have the authors made all data underlying the findings in their manuscript fully available?

Reviewer #1: No

Reviewer #2: No

4. Is the manuscript presented in an intelligible fashion and written in standard English?

Reviewer #1: Yes

Reviewer #2: Yes

5. Review Comments to the Author

Reviewer #1: This paper investigates the use of machine learning to predict adherence to gamified cognitive training programs based on early game performance data. It analyzes baseline characteristics, cognitive game performance, and their predictive power regarding adherence over a ten-week period. The study employs various machine learning algorithms and ensemble modeling techniques to predict adherence, identifying key game performance indicators and their correlation with sustained engagement.

Strengths:

- Innovative approach: The study addresses a significant gap in predicting adherence to cognitive training programs using machine learning techniques, providing insights into the potential of early game performance data.

- Comprehensive analysis: By considering both baseline characteristics and game performance metrics, the study offers a holistic view of factors influencing adherence, enhancing the validity of the predictive models.

- Practical implications: The findings have practical implications for developing tailored adherence promotion strategies, potentially improving the effectiveness of cognitive training programs in older adults.

Weaknesses:

- Sample size and generalizability: The study acknowledges the relatively small sample size and gender imbalance, which may limit the generalizability of the findings, particularly to other demographic groups.

- Data accuracy concerns: The paper acknowledges potential inaccuracies in game play data, which may affect the reliability of the results. Addressing these accuracy issues could strengthen the validity of the findings.

- Limited discussion on recent related works: As a journal article, the paper lacks a thorough review on recent related works concerning machine-learning-derived predictors of cognitive and game performance (e.g., "SPARE-Tau: A flortaucipir machine-learning derived early predictor of cognitive decline" in PLOS ONE'23). Integrating such discussions could provide additional context and insights into the current study's contributions.

Reviewer #2: The article by Zhe He and colleagues presents a comparison of multiple prognostic prediction models which estimate future adherence to game-based cognitive training in the elderly. While the aim and approach seem to be plausible at a high level, the manuscript currently lacks many relevant pieces of information which makes it impossible to fully judge the quality of this study.

Major:

1. Definition of adherence: L131-137 make it sound like weekly adherence should yield a number between 0 (no adherence) and 1 (perfect adherence) in steps of 0.2 (for 5 playing days) which are then averaged over 10 weeks (yielding any number between 0 and 1). This would suggest the use of a beta regression, which does not align with the analytic approach presented below (e.g. L189: t-test/chi-squared test or L206 logistic regression). The authors should be more explicit about their dependent variable and its connection to statistical tests/models.

2. Conceptually, I’m also wondering why the authors chose to build a prognostic prediction model where the outcome is a 10-week average based on data of an initial 2-week playing phase. The ultimate goal was described as developing an adherence promotion strategy by means of notifications - which would be most useful if a probability of adherence is predicted for the upcoming day, so at a lag of 1 day. Given the declared aim and that the study has 12 weeks of data available, it seems odd to have a 10-week average as the outcome and not a model that incorporates all data up until the current day to predict adherence for the next day. Perhaps the authors could clarify their analytical choice and explain what a 10-week average of adherence is supposed to reflect.

3. L167: “how many days a person spent on reaching the median level of each game in the first two weeks” - does this mean, participants were able to play through 50% of the content (= median level) in the first two weeks? If so, can we assume that participants saw all levels after a period of 3-4 weeks and spent week 5-12 repeating the same levels over and over again? This might have big implications for adherence over time (see point 2 above), as participants might be much more likely to stop playing after having seen all content?

4. This study has a wealth of variables available, but there is no table that describes the sample and gives an overview of the used data. The authors should provide a table that contains descriptive statistics of their sample, including raw or standardised scores for the battery of cognitive assessments (Raven, Hopkins memory test, etc.) and questionnaires (General Self-efficacy, Technological Self-efficacy, etc.).

5. The text does not sufficiently refer to existing literature. In the introduction there is no reference for L56-63; the entire method section does not have a single reference although there certainly is the need to back up the multitude of employed machine learning models (I’ve also never heard of the “soft voting ensemble modeling technique” L216 before which is even mentioned in the abstract but nowhere cited). There is not a single reference in the discussion section either, so the authors do not embed their findings into other ongoing efforts in the field.

6. The authors should also elaborate on the analytical approach beyond the one sentence that mentions logistic regression, ridge regression, support vector machines, classification trees and random forests. Is it possible to make the code available and provide model outputs/coefficients?

Minor:

7. Ethics statement: IRB number could be mentioned.

8. Data availability statement: “some restrictions apply” is very unspecific and “all relevant data are within the manuscript and its supporting information” is not true as only results are shown but no raw or processed data underlying these figures.

9. Cognitive training trial (L109+): Were the participants able to select the game(s) they wanted to play? Did they play more than one game in one session? Did they play consecutive levels within one game or switched games after one level?

10. Table 4: The text describes the use of a t-test or chi-squared test (L189) but the table displays a correlation. What kind of correlation was used here and what is the scale of the adherence outcome?

11. The results contain a lengthy description of the AUROC and a guide on how to interpret Figure 2 (L224+). For me, this seems to be better split up and moved into the methods section and the figure caption. Similarly, the description of SHAP (L272+) should be moved into the method section and definitely requires a reference.

12. While considering sex/gender as a predictor is certainly important, it is curious that the correlation coefficient in Table 4 appears to be exactly zero to the fourth digit even though sex differences appear to be well described for (certain) cognitive and technical skills as well as in-game behaviour. I was furthermore wondering how the two participants whose sex is unknown were incorporated into the analysis. Does Table 4 show a rank based correlation and the unknown sex is put in the middle between male and female?

13. Line 71, has the intervention under investigation gained popularity or the general research subject? Please also provide a reference for this.

14. Lines 155-158 are a repetition of what is explained earlier on the page.

15. Line 303 - is ‘proportion’ meant instead of ‘percentage’, as written in L299 (when referring to Supply Run)?

16. Page 15 (no line numbers given) - maintaining adequate level of difficulty is ‘essential’ seems like an overstatement for what this study shows, especially given that difficulty was not assessed and certainly varied over the 10-week outcome period. It could also be that, depending on how many levels the game has, more content is merely necessary to prevent boredom from repeating things.

6. PLOS authors have the option to publish the peer review history of their article (what does this mean?). If published, this will include your full peer review and any attached files.

Reviewer #1: No

Reviewer #2: **Yes: **Toivo Glatz

---

## [Author Response · Author response to Decision Letter 0]

11 Jul 2024

Rsponse to PloS One Reviewers

Reviewer #1: 

1. Sample size and generalizability: The study acknowledges the relatively small sample size and gender imbalance, which may limit the generalizability of the findings, particularly to other demographic groups.

### Response: Thank you for your insightful comments regarding the sample size of our study. We acknowledge the limitation posed by the relatively small sample size of 118 participants, which, as noted, may impact the generalizability of our findings across broader populations. However, several key points support the validity and relevance of our results: 1) The data set utilized was derived from a rigorously designed clinical trial, ensuring high-quality and reliable data; 2) Despite the smaller sample size, the statistical methods employed were carefully chosen to maximize the robustness of our findings. We employed advanced machine learning techniques, which are particularly adept at handling complex interactions and patterns even in smaller datasets. Additionally, the use of five-fold cross-validation helps in validating the generalizability of our predictive models within the given sample; 3) We conducted detailed analyses focusing on the individual games within the cognitive training suite, which revealed specific insights into the patterns of adherence and performance. This granularity allows for more precise understanding and application of the findings; 4) Our results are in line with existing literature, providing an additional layer of validation. In the revised manuscript, we also discuss how our findings compare with those of other studies in terms of predictive factors for adherence to cognitive training. We clearly state that these findings should be interpreted as preliminary and highlight the need for further studies with larger and more diverse populations. This acknowledgment reflects a transparent and cautious approach to the interpretation and application of our results. In addition, we suggest potential strategies for future research to address these limitations, such as multi-site collaborations to enhance sample size and diversity, and the integration of our findings into larger meta-analyses.

2. Data accuracy concerns: The paper acknowledges potential inaccuracies in game play data, which may affect the reliability of the results. Addressing these accuracy issues could strengthen the validity of the findings.

### Response: Thank you for highlighting the concerns regarding potential inaccuracies in the gameplay data, which are indeed crucial for the integrity of our study’s conclusions. We recognize the importance of this issue and have taken several steps to ensure the reliability of our findings: 1) Throughout the study, we implemented rigorous data verification protocols to ensure accuracy in the capture and recording of gameplay data. This includes regular checks for anomalies and inconsistencies in the data collected from the gaming interface; 2) To further mitigate the impact of any potential inaccuracies, we employed robust statistical techniques that are less sensitive to outliers and missing data. These techniques help to strengthen the reliability of our findings despite possible inaccuracies in the raw data; 3) We conducted sensitivity analyses to assess how variations in data accuracy could impact our results. This analysis helps to understand the robustness of our conclusions under different scenarios of data imperfection. We have transparently discussed these potential inaccuracies in the manuscript, acknowledging them as a limitation and suggesting how they might affect the study's outcomes. This approach ensures that readers are aware of these factors when interpreting the findings. In the manuscript, we propose specific recommendations for future studies to enhance data collection and accuracy. This includes the use of more advanced tracking technologies and improved user interfaces that can reduce errors in data capture. We are committed to continuously improving data accuracy in future iterations of the study. This includes refining game design and exploring new technologies that offer more precise and reliable data collection methods. By addressing these potential data inaccuracies through multiple layers of checks and balances, we believe the validity of our findings remains robust. However, we appreciate the importance of continually striving to improve data accuracy and welcome further suggestions or collaborations that could assist in this endeavor.

3. Limited discussion on recent related works: As a journal article, the paper lacks a thorough review on recent related works concerning machine-learning-derived predictors of cognitive and game performance (e.g., "SPARE-Tau: A flortaucipir machine-learning derived early predictor of cognitive decline" in PLOS ONE'23). Integrating such discussions could provide additional context and insights into the current study's contributions.

### Response: This article mentioned by reviewer is not related to our paper. Our main goal of our project is how to improve the adherence of the older adults in cognitive training. But this article, provided by reviwer, aimed to successfully detects pathology in the earliest disease stages of cognitive decline. Now we are conducting a systematic review to investigate what methods other scholars used to increase older adults’ adherence to health-related online training. We found many articles indicated that gamfied environment can help older adults engage with long-term online intervention (Blocker et al., 2014; Scase M et al., 2017). But some authors also indicated that we should personalize these training to mazimize their adherence (Blocker et al., 2014). One article published in PLOS One, written by Turunen M et al. (2019), used several cognitive, demographic, lifestyle, and health-related variables to predict older adults’ adherence to computer-based cognitive training, and they found cognitive functioning may affect their adherence, age, education level and sex may not affect their adherence. But in this article, they only conducted bivariate analyses and multivariate analyses to find the important factors and used ZINB model to predict adherence (Turunen M et al., 2019). In contrast, our project incorporates not only these baseline predictors but also game performance data in the first few weeks of the training. This comprehensive analysis provides more robust findings, which can be used to personalize cognitive training programs and improve adherence among older adults. In the revised manuscript, we have added such discussion in the Introduction section.

Reviewer #2: 

Major:

1. Definition of adherence: L131-137 make it sound like weekly adherence should yield a number between 0 (no adherence) and 1 (perfect adherence) in steps of 0.2 (for 5 playing days) which are then averaged over 10 weeks (yielding any number between 0 and 1). This would suggest the use of a beta regression, which does not align with the analytic approach presented below (e.g. L189: t-test/chi-squared test or L206 logistic regression). The authors should be more explicit about their dependent variable and its connection to statistical tests/models.

### Response: In our study, the value of “weekly adherence” can be greater than 1. For example: if one particpant played 7 days in one week and played for over 10 minutes each day, the weekly adherence of this participant would be 7 divided by 5, which is 1.4. Thus the range of weekly adherence could be between 0 and 1.4. Thus, beta regression is not suitable for this case, as beta regression requires the outcome variable between 0 and 1. And in this study the final response variable is “overall adherence”. To calculate the overall adherence, we took an averaged of the 10 weeks of “weekly adherence” values and then got the median value for the overall adherence for all the 118 participants. If a participant’s average 10-week weekly adherence is greater than the median value, the overall adherence will be recorded as “1”, otherwise it will be recorded as “0”. Thus, the overall adherence is a binary variable. As such, logistic regression is a suitable choose for our situation.

2. Conceptually, I’m also wondering why the authors chose to build a prognostic prediction model where the outcome is a 10-week average based on data of an initial 2-week playing phase. The ultimate goal was described as developing an adherence promotion strategy by means of notifications - which would be most useful if a probability of adherence is predicted for the upcoming day, so at a lag of 1 day. Given the declared aim and that the study has 12 weeks of data available, it seems odd to have a 10-week average as the outcome and not a model that incorporates all data up until the current day to predict adherence for the next day. Perhaps the authors could clarify their analytical choice and explain what a 10-week average of adherence is supposed to reflect. 

### Response: This paper is the continuation of our published work in Information Processing & Management (https://pubmed.ncbi.nlm.nih.gov/35909793/), where we used baseline measures (demographic, attitudinal, and cognitive ability variables) to predict overall 12 week’s adherence. In that paper, we found that we could only achieve an AUROC of 0.71 with the best performing machine learning model such a prediction. So in the current study, we aim to predict 10-week overall adherence using the initial 2-weeks of game play data and the baseline characteristics to provide a more stable and reliable prediction of long-term engagement, smoothing out daily variations. This approach helps in identifying consistent adherence patterns, which are crucial for developing robust, just-in-time intervention strategies. While daily predictions might offer more granular insights, the 10-week average offers a broader perspective that informs sustained engagement and practical intervention planning. The study's key findings demonstrated that game performance in the first two weeks was a superior predictor of adherence compared to baseline characteristics, with games like “Supply Run,” “Ante Up,” and “Sentry Duty” being significant indicators. This method ensures that the predictions are robust and generalizable, thus supporting the development of effective adherence promotion strategies over the long term. Further, we believe that the development of a just-in-time adherence support system that capitalizes on both short-term (day-to-day) and long-term (overall adherence over the remainder of the intervention) predictions might be even more effective than a system that relies solely on short-term predictions. Specifically, the reminder system can be fine-tuned based on the combined forecasts. For example, if the long-term model predicts a high risk of dropout and the short-term model indicates a high probability of non-engagement over the next few days, this might present a more serious concern compared to scenarios where the short-term model predicts an adherence lapse but the long-term model predicts overall high adherence for the remainder of the trial. In such cases, the system might deploy more engaging, context-aware reminders that address specific barriers to long-term adherence to reduce the risk of attrition while also supporting short-term reengagement. We have modified the paper accordingly.

3. L167: “how many days a person spent on reaching the median level of each game in the first two weeks” - does this mean, participants were able to play through 50% of the content (= median level) in the first two weeks? If so, can we assume that participants saw all levels after a period of 3-4 weeks and spent week 5-12 repeating the same levels over and over again? This might have big implications for adherence over time (see point 2 above), as participants might be much more likely to stop playing after having seen all content?

### Response: No. In the variable, “median level” is based on the distribution of the highest levels the participants achieved in the game in the previous two weeks. So if one participant reached the “median level” in previous two weeks, it means that he or she played relativley better than others. And based on our experience, it is almost impossible for the participants to reach the highest level of the games in Mind Frontier. We have made the clarification as “A participant is considered to have surpassed the median level threshold if their maximum level in the game exceeds the 25th percentile of the highest level across all participants, which means that the participant reached a relatively higher level than other participants in previous two weeks.”

4. This study has a wealth of variables available, but there is no table that describes the sample and gives an overview of the used data. The authors should provide a table that contains descriptive statistics of their sample, including raw or standardised scores for the battery of cognitive assessments (Raven, Hopkins memory test, etc.) and questionnaires (General Self-efficacy, Technological Self-efficacy, etc.).

### Response: We have added Table 1 for the descriptive statistic of the cohort of 118 participants in the study. In this dataset, there are 18 z-scores for individual measures and 8 composite scores that combine these measures. Table 4 provides 8 composite scores and we have added two new columns to show the mean and standard deviation of these composite scores (see Table 4). This suggests that there is no significant overall deviation, indicating homogeneity in technology proficecy and self efficiacy within the group. For a comprehensive overview of Attitude and Cognitive Scores for All Participants, please refer to S3 Table.

5. The text does not sufficiently refer to existing literature. In the introduction there is no reference for L56-63; the entire method section does not have a single reference although there certainly is the need to back up the multitude of employed machine learning models (I’ve also never heard of the “soft voting ensemble modeling technique” L216 before which is even mentioned in the abstract but nowhere cited). There is not a single reference in the discussion section either, so the authors do not embed their findings into other ongoing efforts in the field.

### Response: We have added two citiations for L56-63.

“As people age, they are more likely to experience chronic diseases, which can have significant impacts on their quality of life and their ability to live independently. Mild cognitive impairment (MCI), Alzheimer's disease (AD), and related dementias are three of the most common chronic illnesses that lead to memory loss and deterioration of cognitive skills (Kelley & Petersen, 2007). Even for older adults without MCI, AD, or dementia, their cognitive abilities may decline as they age (Mather, 2010). Thus, there is a growing need for care services to support older adults, including medical care, long-term care, and social services. At present, the treatment of memory loss is mainly divided into pharmacological therapy and non-pharmacological therapy.”

About the soft voting ensemble modeling, I found references as the following shows:

“Ensemble modeling is a machine learning technique that combines many models, known as estimators, to create predictions by allowing the models to vote for the predicted class. There are two categories of voting classifiers based on the method of voting: soft and hard voting (Kumari et al., 2021), which are also called unanimous and majority voting (Gandhi & Pandey, 2015). In hard voting, the final class prediction is decided through a majority vote from many estimator models. Soft voting involves the calculation of the probability for each class by each estimator in order to make the final class prediction (Kumari et al., 2021).”

“This classifier is a meta-classifier for merging same or conceptually dissimilar machine learning models for prediction through majority voting. A voting classifier uses two types of voting techniques, hard and soft. In hard voting, the final prediction is done by a majority vote in which the aggregator selects the class prediction that comes again and again among the base models. In soft voting, base models should ha

---

## [Decision Letter · Decision Letter 1]

27 Aug 2024

PONE-D-24-08688R1Predicting Adherence to Gamified Cognitive Training Using Early Phase Game Performance Data: Towards a Just-In-Time Adherence Promotion StrategyPLOS ONE

Dear Dr. He,

Thank you for submitting your manuscript to PLOS ONE. After careful consideration, we feel that it has merit but does not fully meet PLOS ONE’s publication criteria as it currently stands. Therefore, we invite you to submit a revised version of the manuscript that addresses the points raised during the review process.

The Revised Article has been re-reviewed. I agree that some aspects still need revision or clarification. I encourage Authors to take into account the Reviewer's comments.==============================

We look forward to receiving your revised manuscript.

Kind regards,

Stefano Triberti, Ph.D.

Academic Editor

PLOS ONE

Reviewers' comments:

Reviewer's Responses to Questions

**Comments to the Author**

1. If the authors have adequately addressed your comments raised in a previous round of review and you feel that this manuscript is now acceptable for publication, you may indicate that here to bypass the “Comments to the Author” section, enter your conflict of interest statement in the “Confidential to Editor” section, and submit your "Accept" recommendation.

Reviewer #2: (No Response)

2. Is the manuscript technically sound, and do the data support the conclusions?

Reviewer #2: Partly

3. Has the statistical analysis been performed appropriately and rigorously? 

Reviewer #2: No

4. Have the authors made all data underlying the findings in their manuscript fully available?

Reviewer #2: Yes

5. Is the manuscript presented in an intelligible fashion and written in standard English?

Reviewer #2: Yes

6. Review Comments to the Author

Reviewer #2: Thank you for the opportunity to re-review the article by Zhe He et al. While the reporting has improved and the aim and results are more comprehensible now, additional clarifications are required. I would also like to point out that while I understand the general idea behind the soft voting ensemble modelling technique now, I cannot judge whether it has been implemented and interpreted appropriately.

1) Thank you for clarifying the definition of adherence as a median split of the 10-week playing average and the way the binary outcome of the logistic regression is created. I understand that you are doing this split twice for different criteria, one at full adherence and once at minimal adherence which essentially yield 4 different subsamples which form the basis of all analyses. Please expand Table 1 and provide descriptive statistics for these 4 groups (age, sex, weekly adherence and the outcomes stated in Table 3).

2) Relating to point 3 in my previous review report: the manuscript (L195) states that the median level is determined by the “highest levels reached” by all the participants and goes on to explain that if an individual passes the 25th percentile, they played further than 50% of other participants. Shouldn’t this be the 50th percentile (the median?) and if not, could you please provide a clearer description of this procedure?

3) Relating to point 4 in my previous review report: Table 4 does not show new columns with means and standard deviations. Did the authors provide the correct table?

4) Relating to points 10 and 12 in my previous review report: despite the author's response, no changes were implemented in the manuscript. L215 continues to describe t-tests and chi-squared tests and the criterion of a p-value <= 0.1 for including them as predictors. However, Table 5 depicts these as “correlations”. If I compute a chi-squared test on the provided contingency table for minimal adherence, then I get a non-1 p-value, which would suggest some (non-null) differences in the distribution of gender. The authors also comment on Cramer’s V, which is not mentioned in the paper. I continue to be puzzled as to what Table 5 is actually showing. Furthermore, if gender indeed includes the stratum of 2 unknowns, then a chi-squared test cannot be used because the counts within the cells become too small. Fisher’s exact test should be used in such a case.

Minor:

5) It is much appreciated that the authors are sharing the code together with example data. The Notebook for the prediction models appears empty though - APPT_Feb16_24_ClassificationModels.ipynb has zero lines and 2 bytes file size.

6) Table 1. “Std. of Age” is not the common abbreviation of standard deviation (SD). Please explain your abbreviation as part of the table caption/footnote.

7) Table 5 and the line of text above are the only place referring to the variable “condition” - what is meant by this?

8) Legibility of Table S3 would also benefit from rounding to one or two decimals and using a more common formatting such as “mean (SD)” within one cell.

7. PLOS authors have the option to publish the peer review history of their article (what does this mean?). If published, this will include your full peer review and any attached files.

Reviewer #2: **Yes: **Toivo Glatz

---

## [Author Response · Author response to Decision Letter 1]

8 Sep 2024

Response for Reviewer’s Comments

Thanks for the constructive feedback on our manuscript! We have revised the paper point-by-point as follows:

Reviewer #2: Thank you for the opportunity to re-review the article by Zhe He et al. While the reporting has improved and the aim and results are more comprehensible now, additional clarifications are required. I would also like to point out that while I understand the general idea behind the soft voting ensemble modelling technique now, I cannot judge whether it has been implemented and interpreted appropriately.

1) Thank you for clarifying the definition of adherence as a median split of the 10-week playing average and the way the binary outcome of the logistic regression is created. I understand that you are doing this split twice for different criteria, one at full adherence and once at minimal adherence which essentially yield 4 different subsamples which form the basis of all analyses. Please expand Table 1 and provide descriptive statistics for these 4 groups (age, sex, weekly adherence and the outcomes stated in Table 3).

Response: Thank you for the comments to provide me a new angle to make the descriptive statistics more completed. We revised the Table 1. We also added adherence information in Table 3:

2) Relating to point 3 in my previous review report: the manuscript (L195) states that the median level is determined by the “highest levels reached” by all the participants and goes on to explain that if an individual passes the 25th percentile, they played further than 50% of other participants. Shouldn’t this be the 50th percentile (the median?) and if not, could you please provide a clearer description of this procedure?

Response: Sorry for the confusion. To make it more understandable, we changed the variable name from “The number of days reaching median level” as “number of days reaching the middle level group”. In defining the variable 'number of days reaching the middle level group,' we categorized participants' maximum levels from various games over the preceding two weeks into three distinct level group: low level group, middle level group, and high level group. These levels were determined using the 25th and 75th percentiles as thresholds. Specifically, scores falling within the 0-25th percentile were classified as “low level group”, those within the 25th-75th percentile as “middle level groups”, and scores exceeding the 75th percentile as “high level groups”. I revised L184-L191 as:

“In defining the variable 'number of days reaching the middle level group,' we categorized participants' maximum levels from various games over the preceding two weeks into three distinct level groups: low, middle, and high. These levels were determined using the 25th and 75th percentiles as thresholds. Specifically, levels falling within the 0-25th percentile were classified as “low level group”, those within the 25th-75th percentile as “middle level group”, and levels exceeding the 75th percentile as “high level group”. “Number of days reach middle level group” represents the number of days within a two-week span that a participant reaches this middle level group. If a participant does not achieve the middle level group within these two weeks, this variable is set to 15 days.”

3) Relating to point 4 in my previous review report: Table 4 does not show new columns with means and standard deviations. Did the authors provide the correct table?

Response: In our previous submission, we thought the composite score is not raw data, so we provided the means and standard deviations of raw data in S3. Based on your comments, we think it is also necessary to provide the means and standard deviations of composite scores. Thus, we revised the Table 4.

4) Relating to points 10 and 12 in my previous review report: despite the author's response, no changes were implemented in the manuscript. L215 continues to describe t-tests and chi-squared tests and the criterion of a p-value <= 0.1 for including them as predictors. However, Table 5 depicts these as “correlations”. If I compute a chi-squared test on the provided contingency table for minimal adherence, then I get a non-1 p-value, which would suggest some (non-null) differences in the distribution of gender. The authors also comment on Cramer’s V, which is not mentioned in the paper. I continue to be puzzled as to what Table 5 is actually showing. Furthermore, if gender indeed includes the stratum of 2 unknowns, then a chi-squared test cannot be used because the counts within the cells become too small. Fisher’s exact test should be used in such a case.

Response: Sorry for the confusion. We have revised the table as p-values obtained from t-tests. Regarding the concern about applying chi-square tests to assess the correlation of gender and the adherence level, we decided to combine Male and Unknown in one category as “Male & Unknown”. As such, we could use 2x2 contingency table with “Female” and “Male & Unknown” to get the p-value. We have revised the statement in L204-207 and the Table 5 as below:

“To assess the relationship between both continuous and categorical variables with adherence outcomes, t-tests were conducted for continuous variables to determine if their mean values differed significantly between participants who met the adherence threshold and those who did not. Additionally, Chi-square tests were applied to categorical variables to evaluate whether significant associations existed between the categorical variables and adherence. These analyses helped identify key predictors associated with adherence based on both continuous and categorical predictors. We chose those measures that showed statistical significance (alpha = 0.1) in the prediction models, and we ran separate analysis on both minimal level adherence and full level adherence to choose different sets of predictors. To mitigate the potential reduction in validity due to the small sample size in the “Unknown” gender category, we consolidated the gender variable into two groups: 'Female' and 'Male and Unknown.' This adjustment allowed us to apply the Chi-Square test.”

Minor:

5) It is much appreciated that the authors are sharing the code together with example data. The Notebook for the prediction models appears empty though - APPT_Feb16_24_ClassificationModels.ipynb has zero lines and 2 bytes file size.

Response: We don’t know why it was a blank document. We have added the new one. Thank you for the reminder.

6) Table 1. “Std. of Age” is not the common abbreviation of standard deviation (SD). Please explain your abbreviation as part of the table caption/footnote.

Response: We have changed the Table 1 as I presented in previous answer. I changed “Std. of Age” as SD.

7) Table 5 and the line of text above are the only place referring to the variable “condition” - what is meant by this?

Response: The “condition” means that the message condition. During the cognitive training trial, the positive- and negative- framed messages about brain health were delivered to participants via the software program. There were two types of text messages. “1” means that they received positive-framed message (e.g. One of positive-framed message is “Regular metal challenge can have a positive impact on the brain”), “-1” means that they received negative-framed message, (e.g. One of negative-framed message is “Infrequent mental challenge can have a negative impact on brain”)), and “0” means that they received 0 message. We have changed the variable name to “Test Message Reminder Type” and added the description of this variable in Table 3.

To clarify it we added some statement in “The Cognitive Training Trail” part as below:

“The dataset used in this research was obtained from a previous clinical trial conducted on the Mind Frontiers mobile-based cognitive training game suite, designed explore how different message framings (positive-framed and negative-framed) about brain health influence adherence to a technology-based cognitive intervention and to identify individual differences that predict adherence. Florida State University Institutional Review Board approved the study protocol (IRB #: 2017.20622) and informed consent form. The recruitment period of this study was between July 1, 2017, and March 1, 2018. All the recruited participants provided written consent with their signatures. No minors were included in the study. The trial recruited 118 older adults living in the community, with a mean age of 72.6 years and a standard deviation of 5.54. Among these older adults, 78 are female (66%), 38 are male (32%), and 2 are unknown (2%) (see Table 1). Participants were randomly assigned to one of three groups: no message, positive-framed messages, or negative-framed messages (see Table 1). Those in the positive-framing group received messages emphasizing the benefits of engaging with cognitive training (e.g., " Regular metal challenge can have a positive impact on the brain "), while those in the negative-framing group received messages highlighting the risks of not engaging in such training (e.g., " Infrequent mental challenge can have a negative impact on brain "). Participants were instructed to engage in a cognitive training program consisting of seven games (see Table 2), with a target of five sessions per week, each lasting 45 minutes. The data collected included game levels achieved, ranging from 1 to 58, and five possible outcomes: Defeat, Stalemate, Victory, Abort, and Not Yet Finished. The dataset consists of two phases: in Phase 1, participants adhered to a structured 12-week program (45 minutes per day, 5 days per week), while in Phase 2, they were asked to play as frequently as desired during an unstructured 6-week period.”

8) Legibility of Table S3 would also benefit from rounding to one or two decimals and using a more common formatting such as “mean (SD)” within one cell.

Response: We revised Table S3 as you suggested.

---

## [Editor Report · Decision Letter 2]

17 Sep 2024

Predicting Adherence to Gamified Cognitive Training Using Early Phase Game Performance Data: Towards a Just-In-Time Adherence Promotion Strategy

PONE-D-24-08688R2

Dear Dr. He,

We’re pleased to inform you that your manuscript has been judged scientifically suitable for publication and will be formally accepted for publication once it meets all outstanding technical requirements.

Kind regards,

Stefano Triberti, Ph.D.

Academic Editor

PLOS ONE
---

## [Editor Report · Acceptance letter]

24 Sep 2024

PONE-D-24-08688R2 

PLOS ONE

Dear Dr. He, 

I'm pleased to inform you that your manuscript has been deemed suitable for publication in PLOS ONE. Congratulations! Your manuscript is now being handed over to our production team.

Kind regards, 

on behalf of

Prof. Stefano Triberti 

Academic Editor

PLOS ONE